# Timing of Ketogenic Dietary Therapy (KDT) Introduction and Its Impact on Cognitive Profiles in Children with Glut1-DS—A Preliminary Study

**DOI:** 10.3390/children10040681

**Published:** 2023-04-03

**Authors:** Martina Barthold, Anne Jurkutat, Regina Goetz, Lucia Schubring, Juliane Spiegler, Ann-Sophie Fries, Lucia Kiesel, Joerg Klepper

**Affiliations:** 1Department of Special Education and Therapy in Language and Communication Disorders, Julius Maximilians University, 97074 Wuerzburg, Germany; 2Department of Neuropediatrics, Children’s Hospital Aschaffenburg-Alzenau, 63739 Aschaffenburg, Germany; 3Department of Neuropediatrics and Social Pediatrics, Children’s Hospital and Polyclinic Wuerzburg, 97080 Wuerzburg, Germany

**Keywords:** Glut1DS, ketogenic dietary therapy (KDT), cognitive profile, Wechsler intelligence scale (WISC-IV), speech motor impairment, movement disorder

## Abstract

The aim of this research was to characterize cognitive abilities in patients with Glut1-Deficiency syndrome (Glut1DS) following ketogenic diet therapy (KDT). Methods: The cognitive profiles of eight children were assessed using the Wechsler Intelligence Scale (WISC-IV). The effect of ketogenic diet therapy (KDT) on individual subareas of intelligence was analyzed considering the potential influence of speech motor impairments. Results: Patients with Glut1DS showed a wide range of cognitive performance levels. Some participants showed statistically and clinically significant discrepancies between individual subdomains of intelligence. Both variables, KDT initiation as well as duration, had a positive effect on the overall IQ score. Significant correlations were partially found between the time of KDT initiation and the level of IQ scores, depending on the presence of expressive language test demands of the respective subtests of the WISC-IV. Accordingly, the participants benefited les in the linguistic cognitive domain. The discrepancies in cognitive performance profiles of patients with Glut1DS can be attributed to the possibility of a negative distortion of the results due to the influence of speech motor impairments. Conclusions: The individual access skills of test persons should be more strongly considered in test procedures for the assessment of intelligence to reduce the negative influence of motor deficits on test performance. Specific characterization and systematization of the speech disorder are indispensable for determining the severity of speech motor impairment in Glut1DS. Therefore, a stronger focus on dysarthria during diagnosis and therapy is necessary.

## 1. Introduction

Glut1DS is a rare inherited neurometabolic disease. Recent years have shown the increasing complexity of this entity, especially in adolescents and adults. Accordingly, educational, psychological, and linguistic communicative features remain largely unknown. Glut1DS is caused by impaired glucose transport into the brain facilitated by glucose transporter type 1 (Glut1), encoded by *SLC2A1*. As glucose is the essential fuel for the brain, this defect results in a “cerebral energy crisis” in the developing brain, causing epilepsy, developmental delay, and complex movement disorders.

Glut1DS is diagnosed based on pathogenic SLC2A1 variants, reduced glucose concentrations in the cerebrospinal fluid in the presence of normal glucose levels in blood (hypoglycorrhachia), and a suggestive phenotype [1]. First described in 1991, the clinical features of the disorder are increasingly complex and age-specific [2,3]. The disorder is characterized by variable manifestations of a global psychomotor developmental disorder, seizure-like manifestations, microcephalus, and complex movement disorders [1,3,4,5]. The spectrum of impaired movement includes persistent movement disorders and paroxysmal manifestations, such as exercise-induced dyskinesia or spastic choreoathetosis [6], and is triggered by the influence of stressors [7,8,9].

Onset is usually marked by infantile seizures and abnormal eye–head movements. As development progresses, global developmental delay and movement disorders (spasticity, ataxia, dystonia, etc.) develop [1,7,10]. Dysarthria has been reported in the majority of patients [4,6,7,9,11,12,13,14] and may be accompanied by speech incoherence [6] and interfere with the intelligibility of speech [4].

Varesio et al. [15] showed that the presence of movement disorders significantly affects school performance and health aspects, and negatively influences the quality of life of those affected. Klepper et al. [1] also attributed a limited quality of life to motor and speech motor deficits, among other factors.

Cognitive impairment in Glut1DS ranges from learning disorders to severe intelligence deficits [6]. The extent of impairment correlates with the overall severity of the disease [3,8,16]. Average cognitive profiles tend to be exclusive to patients with mild disease [6,14].

Based on the analysis of various indices in the context of intelligence measurements using neuropsychological tests, conclusions can be drawn regarding correlations between the different cognitive components based on performance discrepancies between the individual sub-scores. Single studies have focused on the specific analysis of cognitive profiles, particularly with respect to the selected subdomains of intelligence within the phenotypic spectrum of Glut1DS [12,14,17,18,19,20]. Study results indicate the superiority of linguistic cognitive competencies over cognitive subdomains or overall IQ score [12,14,19,20]. Linguistic receptive performance has been shown to be better preserved when compared to expressive performance [6,9,14]. Several studies also reported developmental delay or an impairment of language competence in the context of global developmental disorders and associated cognitive impairments [12,20]. Zanaboni et al. [20] determined a combination of language and speech motor deficits of various degrees of expression and severity. As such, Glut1DS is considered “a multilevel condition affecting cognitive, motor, speech, and language competencies” (p. 10).

The gold standard of treatment for Glut1DS is ketogenic dietary therapies (KDT), but a lack of efficacy of KDT in late childhood and adulthood has been a controversial issue [13,14,21,22,23,24,25].

The efficacy of KDT in individual cognitive performance domains has not been studied in detail. Preliminary evidence reports an increase in age-related total IQ score at the time of KDT initiation and on KDT duration [11,12,13,14,21].

De Giorgis et al. [12] compared cognitive profiles of 14 subjects with Glut1DS before/after treatment with KDT. Using the Wechsler Intelligence Scale-III [26,27], the authors concluded that the higher the CSF/blood glucose ratio, the longer the duration of KDT, and the lower the age at the time of KDT initiation, the more the patients benefited from it.

Other studies also reported improvements in language skills, e.g., [13,14,28]. In general, the influence of therapy on linguistic competencies in the studies was primarily assessed within the framework of neuropsychological test procedures, and no differentiation was made between the linguistic receptive and linguistic expressive test components. The possible influence of speech motor competencies or limitations of the test persons on test performance has not been considered in any of the studies so far. Systematic investigations of specific linguistic domains and speech motor skills in Glut1DS are still lacking.

The present study aimed to investigate and describe in detail the cognitive performance status of eight children and adolescents with Glut1DS regarding the initiation and duration of KDT on cognitive performance profiles assessed using the Wechsler Intelligence Scale (WISC-IV) [29]. As “measures of general IQ have limited sensitivity to qualitatively different cognitive changes, since they reflect the combined effects of various composite cognitive abilities” [14] (p. 114), individual subdomains of intelligence were specified. In addition, the influence of KDT on the overall intelligence quotient was examined, and the data were analyzed to derive language- and speech-specific features fundamental for an in-depth, disease-specific analysis of Glut1DS.

## 2. Materials and Methods

### 2.1. Participants

Based on regular follow-up examinations at the Glut1DS outpatient clinic at Children’s Hospital, Aschaffenburg, from 2009 to 2021, this study represents an investigator-initiated investigation of eight patients diagnosed with Glut1DS (aged 7.2–15.5 years, M = 11.6; SD = 3.14; range age 7.2–15.5 years). Clinical data of the patients are presented in Table 1.

### 2.2. Neuropsychological Assessment

Neuropsychological testing was administered and scored according to standardized instructions at a separate outpatient appointment. All patients completed the Wechsler Intelligence Scale for Children—Fourth Edition [29]. Total IQ and individual partial performance profiles were determined for each patient (see Table 2). Guidelines on the procedure for evaluating data with this examination instrument were followed as explained by Daseking et al. [30]: to confirm the validity of the total IQ score as a reliable measure of general cognitive performance, a homogeneous index profile was used. In the WISC-IV, four independent cognitive subdomains are defined and form index scores: language comprehension (VCI), perceptual logical thinking index (PRI), working memory index (WMI), and processing speed index (PSI). While the index scores are composed of the performance in ten core tests, the IQ scores of the four indices are included in the total IQ score. If the total IQ score is based on a heterogeneous performance profile, that is, if the difference in the scores in the indices exceeds the threshold, it cannot be considered a reliable measure of overall cognitive performance and should be replaced by the index scores and accompanying analysis of individual differences. Accordingly, the index values can be interpreted as the best characteristic values of specific performance in the respective subareas of intelligence.

### 2.3. Statistical Analyses

The patient profiles in our study with heterogeneous characteristics were subjected to a discrepancy analysis to analyze individual strengths and weaknesses in cognitive subareas. In addition to the analysis of quantitative and qualitative data (Table 1), the potential influence of the dysarthric disorder component on the (linguistic) cognitive performance profile was considered by examining each subtest of the WISC-IV regarding its degree of linguistic expressive demands. Next, the correlation between the variables time of initialization of dietary measures and cognitive performance, as well as duration of treatment and results of individual cognitive functions, especially speech cognitive functions, were evaluated. Correlation analyses of all surveyed parameters were performed according to the results of the normal distribution test using Spearman’s rank correlation.

## 3. Results

### 3.1. Cognitive Performance Profiles in Glut1DS

Retrospective analysis of cognitive performance profiles assessed with the WISC-IV in eight subjects with Glut1DS confirmed the heterogeneity of the cognitive abilities of the patients in interindividual comparison to the age norm [29], expressed by total IQ (TIQ) (see Figure 1—Profil IQ total and indices of the WISC-IV).

Homogeneous index profile (*n* = 5): in the intraindividual comparison, the calculated total IQ value of five subjects (P3, P5, P6, P7, P8) was based on a *homogeneous* index profile; cognitive linguistic competencies developed in these subjects were directly proportional to other subdomains of intelligence.

Heterogeneous index profile (*n* = 3): Significant differences between the index IQ scores (VCI, PRI, WMI, and PSI) were observed in three subjects (P1, P2, and P4). Many children exhibit relative individual strengths or weaknesses in individual intelligence. The differences found must be tested for statistical and clinical significance in pairs using characteristic values to obtain indications of clinically relevant impairments in the sub-areas of intelligence. Therefore, the results of the three subjects with variability between individual index IQ scores were subjected to discrepancy analysis at the index level in accordance with the test instructions of the WISC-IV. In all three cases, the differences exceeded the critical values. Thus, they proved to be statistically significant and clinically meaningful, and could be classified as heterogeneous.

### 3.2. Correlations to KDT

Due to the heterogeneous index profiles observed and supported by the results of the discrepancy comparisons in 3/8 patients with Glut1DS, the correlation between the variables KDT and total IQ score was uninformative. Thus, the influence of KDT was assessed individually for each subdomain of intelligence represented by the VCI, PRI, WMI, and PSI indices.

The results showed differential effects of the time of treatment initiation and the duration of KDT on total IQ as well as the individual cognitive sub-scores; both variables (KDT initiation and duration) had a positive tendency to affect the overall IQ score, thus supporting the findings of De Giorgis [12]. However, the correlation at the level of *r_S_* = −0.611 proved to be non-significant (Table 3). Significant correlations were partially found between the time of KDT initiation and IQ scores with respect to individual subdomains of intelligence.

For the processing speed index (PSI, Table 3), there was a significant correlation of *r_S_* = −0.749 with *p* < 0.05; both subtests included in the index value correlated significantly with the time of KDT initiation, with *r_S_* = −0.639, *p* < 0.05 (coding) and *r_S_* = −0.855, *p* < 0.01 (symbol search), as shown in Table 4.

The correlation between perceptual logical reasoning index score (PRI, Table 3) and KDT initiation was also significant at *r_S_* = −0.719, *p* < 0.05, although only two of the three subtests correlated significantly with the time of treatment initiation (matrix reasoning: *r_S_* = −0.739, *p* < 0.05; picture concepts: *r_S_* = −0.712, *p* < 0.05; block design: *r_S_* = −0.604, n. s.; Table 5).

The relationship between KDT initiation and the working memory index IQ score (WMI) only showed a trend towards significance (*r_S_* = −0.602, *p* = 0.057, Table 3). One of the two subtests (digit span) yielded a significant correlation with KDT initiation at *r_S_* = −0.673, *p* < 0.05; however, the letter-number sequencing subtest fell significantly short of the significance level at *r_S_* = −0.457 (Table 6).

Compared to the other three index scores, the correlation between treatment initiation of KDT and the verbal comprehension index was the weakest (*r_S_* = −0.440, n. s. Table 3). The correlative evaluations at the subtest level also did not show any significant correlations with KDT initiation (similarities: *r_S_* = −0.565, n. s.; vocabulary *r_S_* = −0.424, n.s.; comprehension: *r_S_* = −0.467, n. s.; Table 7).

The examination of the influence of treatment duration on cognitive performance in our study showed a consistent, albeit somewhat weakened picture: the correlation values of the indices were positive, indicating that a longer treatment duration is associated with positive effects on all cognitive performance domains. However, only the correlation between dietary treatment duration and the symbol search subtest of the processing speed index reached a significance level with *r_S_* = −0.624, *p* < 0.05 (Table 4).

## 4. Discussion

This study aimed to assess the effectiveness of KDT on cognitive function and potential correlations with clinical parameters in patients with Glut1DS. For this purpose, we obtained a detailed characterization of cognitive performance profiles in children and adolescents with Glut1DS based on an analysis of individual cognitive sub-performance areas. We assessed the indications of linguistic abilities, as well as the possible negative impact of speech motor impairments on performance in subtest procedures.

Patients with Glut1DS show a wide range of cognitive performance levels compared to age norms [6,12]. Our results support these findings, with cognitive performance profiles ranging from average to far below the average IQ scores.

In line with previous findings indicating the superiority of perceptual versus expressive language performance in Glut1DS, index scores of the WISC-IV, whose subtests are characterized by high linguistic expressive demands, should be more difficult to master for the patient group than index scores with lower linguistic productive demands. Given the speech motor impairments in Glut1DS, we expected intraindividual heterogeneous cognitive profiles, that is, performance discrepancies between individual index IQ scores obtained with the WISC-IV.

Strong deviations in individual index IQ scores in intraindividual comparisons were found in 3/8 subjects. Discrepancy analysis at the index level to test the differences in performance for statistical significance and clinical relevance confirmed the heterogeneity in the performance profiles. Consequently, index IQ scores can be assumed to be characteristic values with the highest significance for the specific cognitive performance of these patients in the respective subdomains of intelligence and represent interindividual cognitive strength and weakness profiles.

For each of the strength domains, IQ scores significantly exceeded the calculated total IQ score. As the calculated total IQ is based on a heterogeneous index profile and does not adequately represent different performances in individual cognitive domains, it cannot be considered a reliable parameter for these study participants.

The heterogeneous index profile in 3/8 participant indicated intraindividual performance discrepancies in the cognitive profile. As an example, subject 4 showed a comparatively weak performance in the subtest VCI with standardized performance in the subtests WMI and PRI (Figure 2), that is, the total IQ of 83 in subject 4 did not reflect the actual cognitive performance. Rather, the respective IQ index values represent more meaningful parameters of the specific performance.

When interpreting the WISC results, the presence of movement disorders in individuals with Glut1DS must be considered. The WISC partly requires access skills for its subtests, which are only given to a limited extent in children with (speech) motor impairments and can thus lead to a threat to test fairness [31]. For individuals “with physical […] impairments […], it is important not to automatically attribute weak performance on an intelligence test to low intellectual performance when it could indeed be attributed to physical impairments. Depending on the nature of the impairments and the requirements of the subtest, performance […] on a standardized test administration may result in scores that underestimate intellectual performance” [32] (p. 59). Motor deficits present in Glut1DS can thus have a disruptive effect on test performance in subtests with a high level of speech motor demands, since in this case, all answers must be given expressively in speech. The subtests with the highest degree of linguistic dependence and correspondingly high effects of speaking ability on demonstrated performance are present in the WISC-IV in the VCI index. For example, speech motor difficulties and speaking effort could lead to decreased motivation and/or exhaustion earlier on the test. In addition, verbal responses could be shorter and thus reflect only part of the child’s knowledge and negatively bias the results, suggesting the underestimation of the linguistic competencies of individuals with Glut1DS, especially in the index language comprehension, which is characterized by high language boundedness and explains the weak results in this sub-performance domain of the WISC [31].

Against the background of the limitations of the applicability of the WISC in Glut1 patients presented above and the small sample of eight subjects, the statistical calculations support the relationship between the dietary treatment, considering both the timing of KDT initiation and the duration of treatment, and the total IQ score obtained, thus demonstrating the effectiveness of the dietary therapy.

To increase the power of the statistical analyses, the effect of KDT on each individual IQ index value was calculated. We identified significant correlations between the KDT and IQ values of the PSI and PRI. The subjects benefited less significantly in the WMI and VCI indices, as expressed by trended correlations. Although subtest-level correlations at the subtest level also did not reveal significant associations with KDT initiation, they were consistently negative. This finding underscores the opposing directions of the parameters analyzed in each case.

These findings are ostensibly at odds with the results of the De Giorgis et al. [12] study, in which the verbal intelligence quotient (VIQ) was presented as a strength domain. This discrepancy can be attributed in part to the fact that the indices of older versions of the WISC are composed of other subtests, and thus the weighting of language productive demands in the subdomains of intelligence differs. Therefore, comparability with the indices of the WISC-IV [29] is not provided.

## 5. Conclusions

Considering the small sample size and the heterogeneous clinical features (e.g., type of seizures), we conclude that the limited significance of the total IQ score and individual index IQ scores resulting from the variable access skills in patients with Glut1DS requires an alternative methodological approach to intelligence diagnostics for this patient group. Cognitive subdomains should always be considered individually and in consideration of the disability-specific symptomatology in Glut1DS regarding the effects of expressive language and/or hand motor test requirements. Each index domain must be examined for the presence of expressive language test demands to quantify the possible negative influence of speech motor deficits on test performance. Accordingly, in the test domain VCI, as the index with the highest speech component in terms of speech motor requirements, followed by the test domain WMI, the smallest effect of therapy in patients with speech impairment is expected. The results of the correlation analyses confirmed the assumption of the dependence of dietary therapy successes, expressed by IQ scores determined with the WISC, on the influence of speech motor skills in both partial performance areas. In this study, the effects of the KDT became more pronounced with a lower linguistic expressive demand level of the subtests in individual indices: IQ scores of the two indices with low language-binding PRI and PSI correlated significantly with the time of therapy initiation. In this sense, the tendential effect of the duration of KDT on cognitive sub-performance domains was also expressed in weaker performances in language-bound indices compared to indices with a lower influence of speech motor skills.

Consequently, in patients with language disorders, the use of less language-related subtests should be considered. The addition of behavioral observations during testing is advised to reduce the possible influence of motor/coordination deficits. To increase test validity, subjects should be prescreened for speech motor deficits to assess the degree of their influence on the test results. To determine the severity of the speech motor impairment, and consequently the extent of the impairment in the validity of the WISC, a specific characterization and systematization of dysarthria in Glut1DS is indispensable. For this purpose, the test instrument Bogenhausener Dysarthrieskalen (BoDyS) [33], which has been available as a version for children (BoDyS-KiD) [34] since 2020, is particularly suitable.

The hypothetical dominance of speech motor skills as a significant influencing variable in the assessment of communicative abilities in patients with Glut1DS highlights the overall need for a stronger focus on dysarthria in diagnostics and therapy.

Due to the rarity of the disease with a correspondingly low prevalence, this study has relevant limitations, especially the heterogeneous nature (age, seizure condition, mutation) of the overall small sample size. Further studies, which should include a larger population, are necessary to obtain better results of the effects of KDT on cognitive profiles in children with Glut1DS. However, our results provide the starting point for further studies with larger samples focusing on speech motor performance in Glut1DS.

## Figures and Tables

**Figure 1 children-10-00681-f001:**
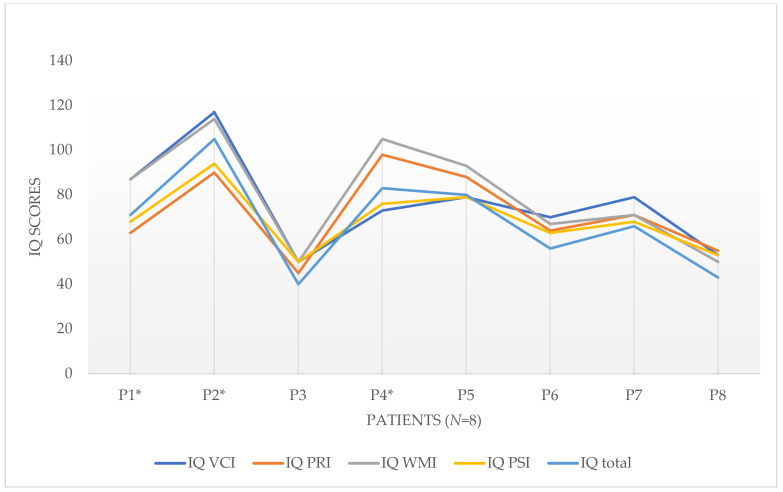
Profil IQ total and indices (WISC-IV), VCI = language comprehension, PRI = perceptual logical thinking index, WMI = working memory index, PSI = processing speed index. * = heterogeneous index profile.

**Figure 2 children-10-00681-f002:**
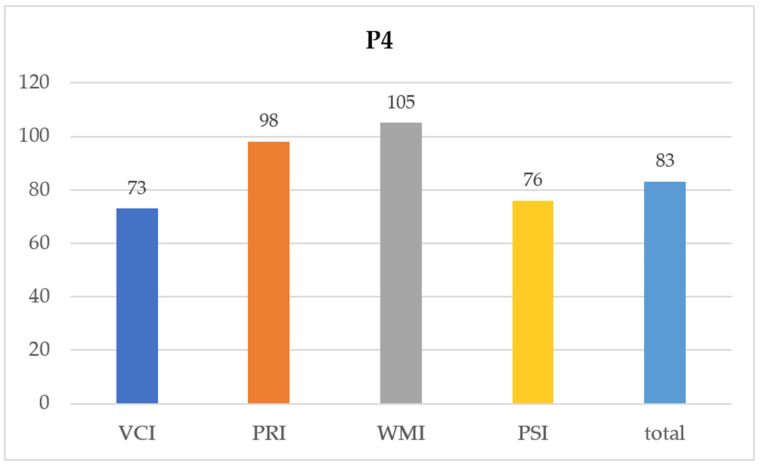
Patient 4 profile—IQ total and indices (WISC-IV).

**Table 1 children-10-00681-t001:** Clinical characteristics.

ID Patient	Sex ^a^	Age at Diagnosis	Mutation	CSF ^b^/ Blood	Epilepsy		Movement Disorder		Other Symptoms ^f^	KDT		KDT Efficacy	
				Ratio	Y/N ^c^	Type ^d^	Y/N	Type ^e^		Start Age	Type ^g^	Epilepsy	Cognition
1	F	5 years	Mut Q304X in Ex7 (nonsense, heterozygous)	/	Y	GS	Y	D, Hy, S, A, PED	ED, SEM	5.2	C	Y	N
2	F	1 year	c.679+2T>G (heterozygous)	0.36	Y	ABS	Y	D, A	OMD	1.0	C	Y	/
3	F	8.5 year	c1199C>T (point mutation)	0.39	Y	ABS, GS, FS	Y	D, Hy, A, PED	H	14.3	C	N	Y
4	F	3 years	R153L (missense, heterozygous)	0.39	N	/	Y	D, A, PED	/	3.1	C, MAD, other	/	/
5	F	3 years	R153L (missense, heterozygous)	0.37	N	/	Y	D, A, PED	/	3.1	C, MAD, other	/	/
6	F	2.5 years	c.26-27insT, Arg11Ser78 (frameshift, heterozygous)	0.33	Y	FS	Y	D, Hy, A, PED	SEM	2.7	C	Y	Y
7	F	3.8 years	c.485T>G(p.(Leu162Arg), Ex 4 (heterozygous)	0.39	Y	ABS, FS	Y	D, A	ED, M	3.8	MAD	Y	Y
8	M	6.6 years	c.138_141dupGACA, (duplicated, heterozygous)	0.37	Y	ABS, FS	Y	D, S, A	/	6.6	C	Y	N

^a^ F = female, M = male. ^b^ CSF = cerebrospinal fluid. ^c^ Y = yes, N = no. ^d^ ABS = absence seizure, FS = focal seizure, GS = generalized seizure. ^e^ A = ataxia, D = dystonia, Hy = hypotonia, PED = paroxysmal exertion-induced dyskinesia, S = spasticity. ^f^ ED = eye deviations, H = hemiplegia, M = myoclonus, OMD = ocular motility disorders, SEM = saccadic eye movements. ^g^ C = classic diet, MAD = modified Atkins diet. / = no data available.

**Table 2 children-10-00681-t002:** Cognitive characteristics.

ID Patient	Sex ^a^	Test	Test Date	Age at the Time of Testing	Dimensions of Intelligence: IQ-Score ^b^				Total IQ- Score
					VC	PR	WM	PS	
1	F	WISC-IV	29.04.16	13.1	87	63	87	68	71
2	F	WISC-IV	20.04.18	8.4	117	90	114	94	105
3	F	WISC-IV	30.09.16	15.5	50	45	50	50	40
4	F	WISC-IV	18.03.16	14.3	73	98	105	76	83
5	F	WISC-IV	18.03.16	14.3	79	88	93	79	80
6	F	WISC-IV	30.08.21	9.1	70	/	67	63	56
7	F	WISC-IV	16.01.20	7.2	79	71	71	68	66
8	M	WISC-IV	25.08.17	10.5	53	55	50	53	43

^a^ F = female, M = male. ^b^ VC = verbal communication, PR = perceptual reasoning, WM = working memory, PS = processing speed.

**Table 3 children-10-00681-t003:** Correlation between cognition (IQ total/indices) and timing of KDT introduction/duration of treatment with KDT.

			Timing of KD Introduction (Age in Years)	Duration of Treatment with KD (in Years)	IQ Total	Verbal Comprehension Index (VCI)	Perceptual Reasoning Index (PRI)	Working Memory Index (WMI)	Processing Speed Index (PSI)
Spearman’s Rho	Timing of KD introduction (age in years)	Correlation Coefficient	1.000	−0.663 *	−0.611	−0.440	−0.719 *	−0.602	−0.749 *
		Sig. (1-tailed)		0.037	0.054	0.138	0.022	0.057	0.016
	Duration of treatment with KD (in years)	Correlation Coefficient		1.000	0.347	0.078	0.383	0.361	0.430
		Sig. (1-tailed)			0.200	0.427	0.174	0.190	0.144

*N* = 8. * Correlation is significant at the 0.05 level (1-tailed).

**Table 4 children-10-00681-t004:** Correlation between subtests of processing speed index (PSI) and timing of KDT introduction/duration of treatment with KDT.

			Timing of KD Introduction (Age in Years)	Duration of Treatment with KD (in Years)	Coding	Symbol Search
Spearman’s Rho	Timing of KD introduction (age in years)	Correlation Coefficient	1.000	−0.663 *	−0.639 *	−0.855 **
		Sig. (1-tailed)		0.037	0.044	0.003
	Duration of treatment with KD (in years)	Correlation Coefficient		1.000	0.253	0.624 *
		Sig. (1-tailed)			0.273	0.049

* Correlation is significant at the 0.05 level (1-tailed). ** Correlation is significant at the 0.01 level (1-tailed).

**Table 5 children-10-00681-t005:** Correlation between subtests of perceptual reasoning index (PRI) and timing of KDT introduction/duration of treatment with KDT.

			Timing of KD Introduction (Age in Years)	Duration of Treatment with KD (in Years)	Block Design	Picture Concepts	Matrix Reasoning
Spearman’s Rho	Timing of KD introduction (age in years)	Correlation Coefficient	1.000	−0.663 *	−0.604	−0.712 *	−0.739 *
		Sig. (1-tailed)		0.037	0.056	0.024	0.018
	Duration of treatment with KD (in years)	Correlation Coefficient		1.000	0.252	0.491	0.533
		Sig. (1-tailed)			0.274	0.108	0.087

* Correlation is significant at the 0.05 level (1-tailed).

**Table 6 children-10-00681-t006:** Correlation between subtests of working memory index (WMI) and timing of KDT introduction/duration of treatment with KDT.

			Timing of KD Introduction (Age in Years)	Duration of Treatment with KD (in Years)	Digit Span	Letter-Number Sequencing
Spearman’s Rho	Timing of KD introduction (age in years)	Correlation Coefficient	1.000	−0.663 *	−0.673 *	−0.457
		Sig. (1-tailed)		0.037	0.034	0.128
	Duration of treatment with KD (in years)	Correlation Coefficient		1.000	0.515	0.235
		Sig. (1-tailed)			0.096	0.288

* Correlation is significant at the 0.05 level (1-tailed).

**Table 7 children-10-00681-t007:** Correlation between subtests of verbal comprehension index (VCI) and timing of KDT introduction/duration of treatment with KDT.

			Timing of KD Introduction (Age in Years)	Duration of Treatment with KD (in Years)	Similarities	Vocabulary	Comprehension
Spearman’s Rho	Timing of KD introduction (age in years)	Correlation Coefficient	1.000	−0.663 *	−0.565	−0.424	−0.467
		Sig. (1-tailed)		0.037	0.072	0.147	0.122
	Duration of treatment with KD (in years)	Correlation Coefficient		1.000	0.319	−0.024	0.224
		Sig. (1-tailed)			0.221	0.477	0.297

* Correlation is significant at the 0.05 level (1-tailed).

## Data Availability

The data presented in this study are available on request from the corresponding author. The data are not publicly available due to privacy of research participants.

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
