# Peer review of "Timing of Ketogenic Dietary Therapy (KDT) Introduction and Its Impact on Cognitive Profiles in Children with Glut1-DS—A Preliminary Study"

_children, 2023, doi:10.3390/children10040681_

Round 1

Reviewer 1 Report

The manuscript entitled “Timing of Ketogenic Dietary Therapy (KDT) introduction and its impact on cognitive profiles in children with Glut1-DS” investigates the effectiveness of KDT on cognitive function and potential correlations with clinical parameters in 8 children with Glut1DS. The study is novel and it demonstrated positive effects of longer KDT duration on cognitive performance domains. However, the major deficiency of the present study is the small sample size (8 subjects) to make a reasonable conclusion. Probably, the number of enrolled individuals is limited to disease rarity (the authors didn’t indicated its frequency and distribution worldwide and in examined population). Therefore, it is hard to conclude on the sufficiency of the analyzed sample size. The results of statistical analysis are logical; however, statistical power is not enough to make pronounced conclusions.

If the Editor decides that this article is suitable for the publication in Children, the authors have to clarify and correct the following issues:

1. It seems to be appropriate to add “a preliminary study” in the title of the manuscript.

2. The Introduction is too extensive (especially, the description of the results reported by De Giorges et al.), please, shorten it.

3. In Materials and Methods section the authors have to indicate whether the diagnosis Glut1DS was made on the basis of ICD/DSM or on sequencing results. Also, there is no information on sequencing results, which are reported in Table 1, in this subsection. Moreover, please, indicate range age of enrolled participants. The information on Bioethical Committee is absent.

4. I would suggest adding standard deviation along to the mean scores in Table 2.

5. In Table 1 please provide the explanation of “/” sign in the table notes.

6. In the Statistical analysis subsection it is unclear why did the authors use Spearman correlation test instead of Pearson correlation test if they have indicated the normality of distribution of cognitive scores? However, based on my experience, the distribution of quantitative trait usually deviates from the normality in such small sample size. Therefore, please provide test statistics to prove that scores of all cognitive domains have been normally distributed.

7. It remains unclear where did the authors have taken the age norm of cognitive abilities (line 201). There is no reference.

8. Together with a correlation analysis performed I suggest to carry out a regression analysis, with normalized by the “age norm” cognitive score (for each cognitive domain) as a dependent variable and all possible factors as potential predictors (for instance, KDT initiation and duration, sex, epilepsy, severity of movement impairments, speech motor deficits, etc.) with subsequent inclusion of only significant predictors in the model.

9. What subjects demonstrated the “Heterogeneous index profile”? (line 222). Please, mark them on Fig.1 and mention  in the text.

10. Please, delete “This section may be divided by subheadings. It should provide a concise and precise 195 description of the experimental results, their interpretation, as well as the experimental 196 conclusions that can be drawn.” (line 195).

11. The authors have to provide the title of Fig. 1 prior to its explanation. Although, the abbreviations given on Fig. 1 have to be explained in its notes.

12. The titles of Tables 4-7 have to indicate the variables for which correlations were calculated. Please, correct the titles analogous to the title of Table 3.

13. Check if the title of Fig. 2. is correct.

14. Finally, the authors have to indicate the limitations of the revealed findings.

Reviewer 2 Report

Introduction

I suggest making shorter the Introduction section.

Methods

-Given that you have a small sample, you should describe the participants deeply.

-I am pretty sure that the type of epilepsy influences cognitive profiles and the effect of KDP. I think that is not the same size for participants suffering from a generalized seizure as those suffering from a focal seizure. How did you control that?

-There is certain coloniality between each subtest and indices. Why did you correlate each one? I suggest choosing only indices or subtests.

-I am still determining if this kind of analysis is correct for this data type, but I suggest changing it.

Results

I suggest editing tables 3, 4, 6, and 7. It would be better to see only the first two rows:  timing of KD introduction (age in years) and duration of treatment with KD (in years) and their correlation with WAIS indices.

Discussion

-I am not sure whether your data may support your conclusions.

-In figure 2, you should eliminate IQ from the bar graph name.

Reviewer 3 Report

Minor issue

- Lines 366-379 and Lines 381-398, it seems that there are two conclusions for this study. Please summarize all the results as well as practical implications based on the results of the study in only ONE Conclusion.

Round 2

Reviewer 1 Report

The authors have made all necessary corrections and reported point-to-point answer on my comments, therefore, the manuscript is now suitable for publication.

Reviewer 2 Report

The paper was modified as I suggested.